# A Genotyping Method for Detecting Foreign Buffalo Material in Mozzarella di Bufala Campana Cheese Using Allele-Specific- and Single-Tube Heminested-Polymerase Chain Reaction

**DOI:** 10.3390/foods12122399

**Published:** 2023-06-16

**Authors:** Rosario Rullo, Simonetta Caira, Ioana Nicolae, Francesca Marino, Francesco Addeo, Andrea Scaloni

**Affiliations:** 1Institute for the Animal Production System in the Mediterranean Environment, National Research Council, 80055 Portici, Italy; simonetta.caira@cnr.it (S.C.); andrea.scaloni@cnr.it (A.S.); 2Research and Development Institute for Bovine, 077015 Balotesti, Romania; ioana_nicolae@yahoo.com; 3Department of Clinical Medicine and Surgery, Endocrinology Unit, University Federico II, 80131 Naples, Italy; francesca.marino2@gmail.com; 4Dipartimento di Agraria, Università degli Studi di Napoli “Federico II”, 80055 Portici, Italy; doglie42@gmail.com

**Keywords:** PDO cheese, water buffalo milk, CSN1S1-deleted allele, PCR analysis, food fraud

## Abstract

Mozzarella di Bufala Campana (MdBC) cheese is a Protected Designation of Origin (PDO) product that is important for the economy and cultural heritage of the Campania region. Food fraud can undermine consumers’ trust in this dairy product and harm the livelihood of local producers. The current methods for detecting adulteration in MdBC cheese due to the use of buffalo material from foreign countries could exhibit limitations associated with the required use of expensive equipment, time-consuming procedures, and specialized personnel. To address these limits here, we propose a rapid, reliable, and cost-effective genotyping method that can detect foreign buffalo milk in a counterpart from the PDO area and in MdBC cheese, ensuring the quality and authenticity of the latter dairy product. This method is based on dedicated allele-specific and single-tube heminested polymerase chain reaction procedures. By using allele-specific primers that are designed to detect the nucleotide g.472G>C mutation of the *CSN1S1B^bt^* allele, we distinguished an amplicon of 330 bp in the amplification product of DNA when extracted from milk and cheese, which is specific to the material originating from foreign countries. By spiking foreign milk samples with known amounts of the counterpart from the PDO area, the sensitivity of this assay was determined to be 0.01% *v*/*v* foreign to PDO milk. Based on a rough estimate of its simplicity, reliability, and cost, this method could be a valuable tool for identifying adulterated buffalo PDO dairy products.

## 1. Introduction

Mozzarella di Bufala Campana (MdBC) is a premium quality cheese made from the milk of the Italian Mediterranean buffalo reared within the EU’s Protected Designation of Origin (PDO) area, which includes the Campania Region and a few geographical districts in Lazio, Molise, and Puglia, as defined by EEC Regulation N° 1107/96. The production of MdBC cheese is a labor-intensive process that has been passed down for generations and supports the livelihood of many local farmers and producers, notably contributing to the Italian agrifood economy. In addition to the problem of maintaining the traditional production process, MdBC cheese is currently facing novel challenges. The augmented demand for this dairy product has led to fraudulent practices involving the illegal addition of cow’s milk to the buffalo counterpart or buffalo material (milk or curd) from foreign countries being added to the buffalo milk from the PDO area. This former adulteration can be assessed using dedicated analytical procedures based on polyacrylamide gel isoelectric focusing, high-pressure liquid chromatography, and the mass spectrometry analysis of caseins [1,2,3,4], whereas the latter represents a more subtle form of fraud, which, in turn, necessitates the development of reliable and efficient detection methods. In the latter context, we have previously demonstrated that Venezuelan, Romanian, Bulgarian, Polish, and Canadian buffalo milk can be distinguished from the Italian Mediterranean counterpart for the specific presence of the β-casein isoform A and an internally deleted αs1-casein variant lacking eight amino acids (f35–42) [3,5]. In dedicated protein-centered studies based on mass spectrometry techniques, these casein isoforms were proposed as specific markers to discriminate buffalo milk from the PDO area and MdBC cheese from counterparts based on or containing buffalo material from foreign countries [3,5]. The above-reported detection methods necessitate expensive equipment, time-consuming procedures, and specialized personnel.

The *CSN1S1* locus in buffalo appears to vary from that of other ruminant species, such as goats, in which a significant degree of polymorphism has been detected, with the identification of 18 allelic variants, which are also associated with quali-quantitative expression differences [6,7]. In Indian buffalo, only a mutated *CSN1S1* allele, featuring the c.620G>A substitution, has so far been characterized [8]. On the other hand, the Romanian buffalo carries the *CSN1S1B^bt^* allele with nucleotide g.472G>C replacement [9]. The latter substitution inactivates the intron 6 splice donor site, resulting in the skipping of exon 6 in the corresponding mRNA and the production of the aforementioned internally deleted protein. The *CSN1S1B^bt^* allele has a frequency of approximately 0.2 in Romanian buffaloes [9]; a similar frequency is also thought to occur in animal populations from Venezuela, Bulgaria, Poland, and Canada. Notwithstanding dedicated studies, no *CSN1S1B* polymorphism has been detected in Mediterranean buffaloes from the PDO area, probably as a result of the conservative aptitude of the local breeders in introducing individuals from other animal breeds [10].

Recent advancements in recombinant DNA technologies have facilitated the detection of casein polymorphisms in buffalo breeds and populations. In this context, polymerase chain reaction (PCR)-based methods have proven to be valuable tools for verifying the authenticity of mono-species dairy products and protecting them from illegal adulterations due to the addition of material from other organisms. Specific applications have focused on detecting undeclared bovine milk in buffalo dairy products [11,12,13]. However, these methods are not optimized to detect mono-species material from undeclared geographical regions in PDO products. The primary objective of the present study was the detection of buffalo material from foreign countries in buffalo milk from the PDO area and MdBC cheese. To accomplish this goal, we developed a dedicated genotyping method that utilized allele-specific PCR (AS-PCR) and single-tube heminested PCR (STHN-PCR) to detect the occurrence of the *CSN1S1B^bt^* allele in the above-mentioned PDO dairy products. Our approach was designed to be rapid, sensitive, and specific, making it ideal for the quality control and authenticity testing of buffalo milk from the PDO area and MdBC cheese.

## 2. Materials and Methods

### 2.1. Milk and Cheese Samples

Bulk milk and cheese samples were collected from the dairy industry, with a daily yield of 900 to 2700 animals. Fresh bulk milk samples (100 mL) were obtained from Romanian (18 in number), Bulgarian (11 in number), and PDO area (40 in number) farms; the number of animals contributing to each sample was unspecified. Fresh milk samples (100 mL) from single animals were obtained from Romanian (17 in number), Bulgarian (17 in number), and PDO area (21 in number) farms. All milk samples were screened for the detection of the α_s1_-casein variant (f35–42) [3,5] and then used to develop and validate the genotyping method reported in the present study. MdBC cheese samples (30 in number) and mozzarella cheese imitation products (10 in number) were obtained from cheese plants located in the PDO area and in eastern European countries, respectively. Fresh milk and cheese samples were collected and stored in individual lots at −20 °C until analysis.

### 2.2. Treatment and Analysis of Milk and Cheese Samples

Genomic DNA was extracted from bulk and single-animal milk from foreign countries and the PDO area, as well as mozzarella cheese from foreign countries and PDO dairies, for a total of 164 samples analyzed. This high number allowed us to obtain a good quality and amount of nucleotide material that was suitable for PCR analysis as well as to characterize the peculiar genotype of each sample group. To recover buffalo somatic cells, we centrifuged 25 mL of raw milk at 2500× *g* for 10 min at 4 °C; the recovered cellular pellet was washed three times with a PBS buffer (0.137 M NaCl, 0.0027 M KCl, 0.01 M Na_2_HPO_4_ and 0.0018 M KH_2_PO_4_, pH 7.4). Then, DNA was extracted from the samples using the commercial Wizard DNA cleanup extraction kit, according to the manufacturer’s instructions (Promega, Madison, WI, USA). For the extraction of DNA from cheese samples, an adapted protocol was used. An aliquot of cheese (approximately 2 g) was homogenized using an Ultra-Turrax disperser (IKA-Werke GmbH, Germany) in 5 mL of a CTAB buffer (100 mM Tris HCl, 20 mM EDTA, 1.4 M NaCl and 2% *w*/*v* cetyltrimethylammonium bromide, pH 8.0). The resulting material was treated with 50 µL of 20 mg/mL proteinase K (Sigma-Aldrich, Milan, Italy) and was incubated overnight under gentle shaking at 55 °C. After centrifugation at 4000× *g* for 15 min, the upper-fat layer was removed, and the resultant DNA was extracted three times with 5 mL of phenol/chloroform/isoamyl alcohol (25:24:1, *v/v/v*) and subsequently with 3.5 mL of chloroform/isoamyl alcohol (24:1, *v/v*). At each step, the solution was centrifuged at 10,000× *g* for 15 min, according to Sambrook and Russell [14]. DNA was then precipitated with 3.5 mL of cold isopropanol, incubated at −20 °C for approximately 30 min, and centrifuged at 10,000× *g* for 15 min. After dissolution in 100 µL of nuclease-free water, DNA was quantified using a Nanodrop ND-2000C spectrophotometer (Thermo Scientific, Milan, Italy) and qualitatively analyzed on a 0.8% *w*/*v* agarose gel using reagents from Merck Sigma-Aldrich (Milan, Italy). 

### 2.3. Design of AS- and STHN-PCR Primers

Allele-specific (AS) and single-tube heminested (STHN) PCR primers were designed on the basis of the genomic sequence of *Bubalus bubalis* (GenBank accession number KJ635888) and the partial CDS of the *CSN1S1B^bt^* allele (GenBank accession number JN786873) using Primer3 software (http://primer3.ut.ee) as follows: A1C-FW (5′-TGC AGC AGC AAA ATG TAA GG-3′), A1C-RV (5′-CTA ATG CCT TGT CTC CTC ACC GTT-3′) and AS-FW (5′-AGA AGG TCA ATG AAC TGA GCA AGC-3′). The former two oligonucleotides were fully compatible with the other gene sequences of *Bubalus bubalis*, which are available in the GenBank database (accession numbers JN786874 and JN786872). The underlined nucleotides indicated the mismatch positions for the *CSN1S1B^bt^* allele-specific amplification. To verify the amplification failure, we used the primer pair of LALBA-FW (5′-GAC CCC ATT TCA GGA TCT TG-3′) and LALBA-RV (5′-TGG CAG ACC ACA GAG TAT CT-3′), which were synthesized based on the α-lactalbumin gene complete CDS (GenBank accession number AF194373.1) and used as an internal positive control. 

### 2.4. AS-PCR and STHN-PCR

To establish the presence/absence of amplified fragments with certainty and to directly determine the sample genotype, the AS-PCR analysis of DNA extracted from the somatic cells of milk samples (see Materials and Methods, Section 2.2) was performed using AS-FW and A1C-RV primers, of which the former was specific for the polymorphic variant. In parallel, the DNA extracted from cheese samples was analyzed using STHN-PCR [15,16] with a set of outer primers A1C-FW and A1C-RV, which were utilized in combination with the internal primer AS-FW. To increase the assay sensitivity and remove carry-over contamination, the subsequent opening of the tubes was avoided by adding reagents and oligonucleotides before the beginning of the amplification reaction [17,18]. AS-PCR and STHN-PCR analyses were performed in a total volume of 50 µL, containing 100 ng of genomic DNA, 0.4 µM of each primer, and a 1x My Taq reaction buffer that included 5 mM dNTPs, 15 mM MgCl_2_, and 2.5 U of MyTaq HS DNA polymerase (Bioline, London, UK). AS-PCR amplifications were carried out in a thermal cycler (MiniCycler PTC-150 MJ Research, Genencor, Life-Science, Milan, Italy) under the following conditions: an initial denaturation step at 95 °C for 1 min, followed by 35 cycles at 95 °C for 15 s, 58 °C for 15 s, and 72 °C for 10 s, and a final extension step at 72 °C for 1 min. For STHN-PCR, the annealing temperature was changed from 58 to 55 °C. These PCR products were analyzed on a 2% *w*/*v* agarose gel (Merck Sigma-Aldrich, Milan, Italy) and stained with ethidium bromide. To confirm the absence of undeclared DNA from foreign buffaloes and to exclude the possibility of false PCR positive results, this reaction was performed with all components of the PCR mix except for the genomic DNA sample. 

### 2.5. Sensitivity of the Genotyping Method

To evaluate the detection limit of the AS-PCR method, a series of amplification assays were carried out on heterozygous DNA at locus *CSN1S1* (*CSN1S1B*/*CSN1S1B^bt^*), which were extracted from single buffalo milk samples from foreign countries. These samples were diluted (*v/v*) with increasing amounts of a homozygous DNA counterpart (*CSN1S1B*/*CSN1S1B*), which was extracted from single buffalo milk samples from the PDO area to reach a heterozygous DNA concentration range from 100 to 0.02 ng; this corresponded to 50% and 0.01% of the *CSN1S1B^bt^* allele, respectively. Gels were assessed based on densitometric analysis using ImageJ 1.42q software v. 1, and the data were reported as the mean ± standard deviation (n = 3). The statistical significance of differences among various dilutions was evaluated using Student’s *t*-test, for which the threshold was *p* < 0.05.

### 2.6. DNA Sequencing Analysis

For DNA sequencing analysis, the homozygous and heterozygous DNA extracted from the milk samples were amplified using the A1C-FW and A1C-RV primers and sequenced using the Sanger method [19].

## 3. Results and Discussion

### 3.1. Detection of Buffalo Milk from Foreign Countries in the Counterpart from the PDO Area

Previous proteomic studies have shown that the internally deleted α_s1_-CN variant (f35–42) is present in buffalo breeds/populations from Venezuela, Romania, Bulgaria, Poland, and Canada [5,10]. Subsequent genetic investigations have confirmed this finding in Romanian animals [9], which has led us to develop a PCR-based method to differentiate between the variant *CSN1S1B^bt^* and the wild-type *CSN1S1B* allele. We achieved this by designing two dedicated primers, namely A1C-RV and AS-FW, of which the latter has the potential to discriminate alleles due to its specificity at the 22nd (G>C) and 24th (C>A) bases. The absence of 3′→5′ exonuclease activity in Taq DNA polymerase used in our experiments meant that the presence of two mismatches in the AS-FW primer had a significant impact on the enzymatic performance of the amplification reaction. Specifically, the mismatches inhibited the polymerase’s ability to amplify the wild-type allele while increasing its specificity toward the mutated allele [20,21].

To test the efficacy of the AS-FW primer in detecting the g.472G>C mutation, we performed AS-PCR on 55 DNA samples extracted from the milk of 17, 17, and 21 individual animals from Romania, Bulgaria, and the PDO area, respectively. The resulting 330 bp amplicon, corresponding to the *CSN1S1B^bt^* allele, was detected only in DNA samples from the milk of foreign buffaloes (Figure 1 and Table 1). These results confirmed the specificity of the AS-FW primer in detecting the mutation associated with milk from foreign animals. Notwithstanding the reduced number of foreign milk samples assayed, we preliminarily verified that the observed frequency of the *CSN1S1B^bt^* allele was roughly equal to the one measured in previous investigations [9]. We also performed comparative genotyping analysis using the DNA from bulk milk samples collected from Romania, Bulgaria, and the PDO area. As shown in Figure 1 and Table 1, we obtained the same results, with the *CSN1S1B^bt^* allele present only in the bulk milk samples from Eastern European countries. This finding confirmed previous studies indicating that the *CSN1S1B^bt^* allele is absent in the buffalo population of the PDO area [5,10]. Positive internal control was included in all AS–PCR reaction mixtures to ensure the reliability of the above-mentioned results. In particular, a specific 385 bp amplicon corresponding to the *LALBA* gene was amplified in all the samples; this allowed us to verify that the absence of the band corresponding to the *CSN1S1B^bt^* allele in the milk samples from the PDO area was not due to possible technical problems. 

To evaluate the sensitivity of the above-reported AS-PCR method, additional experiments were then performed on the milk from single Romanian animals, heterozygous at the *CSN1S1* locus, which was gradually diluted with milk containing homozygous DNA (from the PDO area) to obtain samples with *CSN1S1B^bt^* amounts in the range of 100 to 0.02 ng [22]. As shown in Figure 2A, the intensity of the bands corresponding to the 330 bp amplicon gradually decreased with decreasing amounts of DNA from the foreign milk. This result indicated that the developed AS-PCR method was highly sensitive and could accurately detect even small amounts of foreign heterozygous DNA material in the PCR reaction mixture. Indeed, we detected the presence of the *CSN1S1B^bt^* allele in foreign buffalo milk up to a dilution of 1:5000 (foreign to PDO milk), as shown in lane 8 of Figure 2A. To further quantify the sensitivity of this genotyping method, densitometric analysis was used to measure the intensity of the bands on the stained gels. Using the resulting linear calibration curve, we calculated the limit of detection (LOD) of the *CSN1S1B^bt^* allele to be 0.01% *v*/*v* (foreign to PDO milk), corresponding to about 0.02 ng of the foreign heterozygous DNA (Figure 2B). Additionally, we found that the limit of quantification (LOQ) was 0.6% *v*/*v* (foreign to PDO milk). These results were consistent with those obtained using other DNA-based techniques for species identification in dairy products [23] and demonstrate that the AS-PCR method reported here was highly sensitive and could accurately detect even small amounts of DNA material from foreign buffalo milk in the counterpart from the PDO area. This is crucial for maintaining the quality and authenticity of MdBC cheese and ensuring consumer protection from food fraud. 

Table 2 provides a summary of the differences in performance between the analytical methods developed so far for assessing the adulteration of buffalo milk from the PDO area with a foreign counterpart [5,24]. The present study is shown with details of the corresponding sensitivity, analysis time, instrument cost, and analysis cost in rough values. It is worth noting that previous quantitative approaches based on combined isoelectric focusing and immunoblotting have detected only a minimum of 3% *v*/*v* of foreign buffalo milk in the counterpart from the PDO area. Moreover, the AS-PCR method here reported ensured a very high selectivity and the absence of interferences due to unknown parameters by being exclusively dependent on the generation of a unique allelic-specific amplicon.

### 3.2. Detection of Buffalo Material from Foreign Countries in MdBC Cheese from the PDO Area

Somatic cells in milk, typically present at high levels, provide a good source of DNA for further dedicated analyses. However, various factors, such as raw material processing and environmental conditions, could affect the quantity and quality of extracted DNA, limiting the sensitivity and accuracy of the corresponding PCR analysis [25]. Thus, the careful optimization of DNA extraction protocols was necessary to obtain high-quality DNA for further food adulteration assays. To improve the yield and sensitivity of the PCR analysis performed on MdBC samples and mozzarella imitations made with foreign buffalo milk as well as to prevent non-specific amplification phenomena, a method for dedicated STHN-PCR analysis was developed using two outside primers and an inside counterpart in a single amplification test [17,18,26]. This was conducted because attempts to apply the above-reported AS-PCR method to the analysis of cheese samples showed partial limitations due to the quality of the extracted DNA. The outer primers A1C-FW and A1C-RV were designed to amplify a 410 bp fragment of the *CSN1S1B* gene, which included the g.472 nucleotide position. The inner primer AS-FW was designed to specifically amplify the *CSN1S1B^bt^* allele-specific 330 bp amplicon, as reported above. Accordingly, the amplification of the 410 bp fragment for the identification of buffalo DNA was achieved in all cheese samples made with milk from animals with varying genotype characteristics (Figure 3). This fragment was used as an internal positive control for PCR reactions and the further sequencing analysis of the recovered DNA material. By contrast, the 330 bp fragment was associated with the above-reported *CSN1S1B^bt^* allele-specific amplicon and occurred uniquely in mozzarella imitations made with foreign buffalo milk (Figure 3).

The nucleotide sequencing analysis of the 410 bp fragment from both homozygous and heterozygous DNA is presented in Figure 4. Specifically, Figure 4A displays the nucleotide sequence of the homozygous sample, which revealed there was a G at position 70 and C at position 145, which is in agreement with the genotype characteristics of the Italian Mediterranean buffalo reared in the PDO area. On the other hand, Figure 4B shows the nucleotide sequence of the heterozygous sample, which displayed two ambiguous nucleotide attributions at the above-reported positions. The nucleotide ambiguities found in the heterozygous DNA sample correspond to the presence of the G and C nucleotides already detected in the homozygous DNA at the same positions, which were overlapped by C and T nucleotides from the mutated allele. These observations indicate that the heterozygous sample contained both alleles of the *CSN1S1* gene, namely *CSN1S1B* and *CSN1S1B^bt^*, which could be distinguished by the two single nucleotide polymorphisms (SNPs) G>C and C>T, respectively. On this basis, the identified genomic differences at the *CSN1S1* locus between the buffalo populations from different geographical areas could serve as useful genetic markers for their differentiation, but they could also be instrumental for detecting the use of non-declared material from foreign countries in the preparation of MdBC cheese and other dairy products from the PDO area.

## 4. Conclusions

In addition to confirming the potential of using genetic markers to distinguish geographically separated animal populations, the present study highlights the useful application of related DNA-based technologies to develop valuable tools for the protection of derived raw materials against intra-species food adulterations. Taking advantage of the peculiar condition of the genetic isolation of the buffalo populations reared in Campania, Lazio, Molise, and Puglia, this investigation presents a reliable and sensitive method to ensure the integrity of MdBC cheese and other dairy products from the PDO area, protecting the reputation of the corresponding industry. In fact, this proposed approach could be used to effectively detect the presence of buffalo milk from foreign countries in the counterpart from the PDO area at a detection limit of 0.01% *v*/*v* contamination. The reported method is easy to use, cost-effective, and suitable for routine examinations in Official Control Laboratories. Accordingly, it can be used to effectively screen the authenticity of milk samples from farm sites before using them in cheese manufacturing. Similar DNA-based approaches have already been applied to evaluate animal breeds and plant ecotypes, which are used to produce specific PDO products, ensuring their traceability; this is the case for the bovine breed used for *Ternera de Navarra* beef [27], the fish eggs from the flathead mullet in *avgotaracho Mesolonghiou* products [28], the olive cultivar used in *Terra di Bari* extra virgin olive oil [29], and the tomato ecotype for the peeled product *Agro Sarnese-Nocerino San Marzano Tomato* [30].

The wide variability in the frequency of the internally deleted α_s1_-casein variant in various investigated buffalo breeds/populations and countries could sometimes affect quantitative information regarding extraneous milk spiking into the PDO counterpart. Despite this limitation, the mere presence of foreign buffalo milk, which is evidence against the authenticity of a cheese sample under consideration, can alarm the control authority on the non-declared importation of buffalo material from foreign countries into the PDO area. In conclusion, the proposed DNA-based method is a powerful tool to protect the authenticity of MdBC cheese and the PDO label against fraud and illegal practices. Avoiding the importation of foreign buffalo animals or cryopreserved buffalo semen straws into Italy is also highly recommended to preserve the unique molecular profile of milk and dairy products from the PDO area and ensure their corresponding recognizability. In fact, the long-standing isolation of the Italian buffalo population, with no genetic exchange with foreign animals, has contributed to determining the absence of the casein polymorphism phenomena otherwise detected in other breeds worldwide.

## Figures and Tables

**Figure 1 foods-12-02399-f001:**
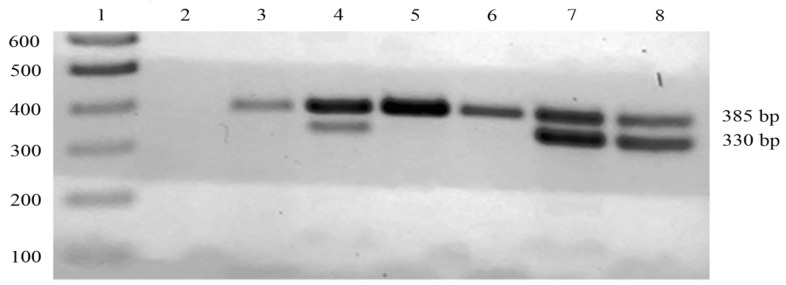
AS-PCR detection of the *CSN1S1B^bt^* allele in buffalo milk from different geographical locations. One-tenth of each AS-PCR product was loaded onto a 2% *w*/*v* agarose gel. Lane 1, 100 bp DNA ladder; lane 2, negative control; lane 3, milk of a single animal from the PDO area; lane 4, bulk milk from Romania; lanes 5 and 6, bulk milk from the PDO area; lanes 7 and 8, milk of a single animal having the *CSN1S1B^bt^* allele from Romania and Bulgaria, respectively.

**Figure 2 foods-12-02399-f002:**
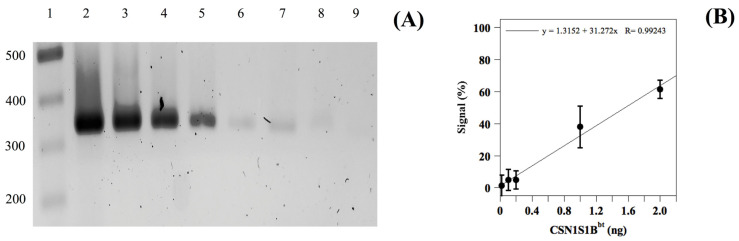
Evaluation of the detection limit of heterozygous DNA (*CSN1S1B*/*CSN1S1B^bt^*) from buffalo milk. Panel (**A**). AS-PCR products of buffalo milk of a single foreign animal containing heterozygous DNA (*CSN1S1B*/*CSN1S1B^bt^*) progressively diluted with the counterpart from the PDO area containing homozygous DNA (*CSN1S1B*/*CSN1S1B*); reaction products were loaded onto 2% agarose gel. Lane 1, 100 bp DNA ladder; lane 2, non-diluted buffalo milk from Romania; lanes 3 to 8, buffalo milk of a single foreign animal progressively diluted (1:10, 1:50, 1:100, 1:500, 1:1000, and 1:5000) with the counterpart from the PDO area, respectively; lane 9, negative control. Panel (**B**). Densitometric analysis of the gel bands 4–8 reported in panel A using ImageJ software. Data are reported as the mean percentage values ± SD with respect to the result from lane 2 (used as 100%). The experiment was performed in triplicate.

**Figure 3 foods-12-02399-f003:**
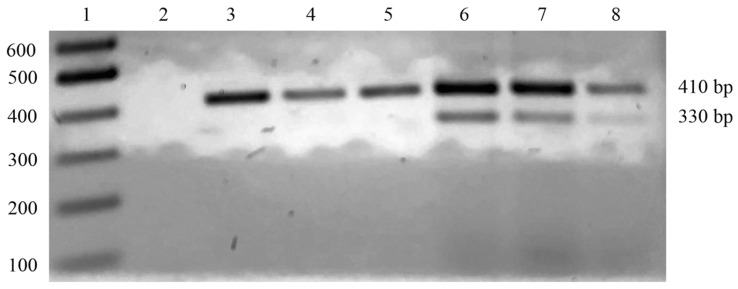
STHN-PCR analysis of DNA from authentic PDO Mozzarella di Bufala Campana samples and mozzarella imitations made with buffalo milk from foreign countries. Three-tenths of each STHN-PCR product was loaded onto a 2% agarose gel. Lane 1, 100 bp DNA ladder; lane 2, negative control; lanes 3–5, PDO MdBC samples from the PDO area; lanes 6–7, mozzarella imitation samples made with buffalo milk from Romania; lane 8, mozzarella imitation sample made with buffalo milk from Bulgaria.

**Figure 4 foods-12-02399-f004:**
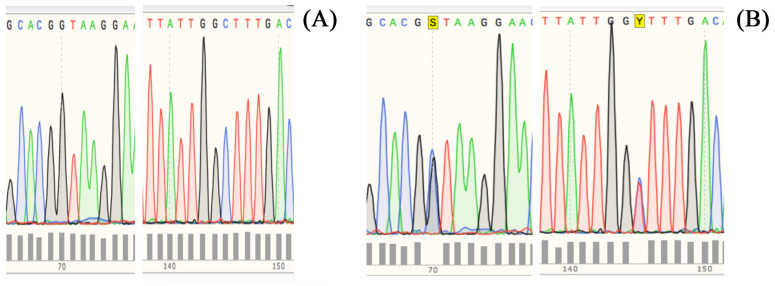
Electropherograms from Sanger sequencing of the 410 bp amplicon from homozygous and heterozygous DNA in identifying polymorphic sites. Panel (**A**). Homozygous DNA (*CSN1S1B*/*CSN1S1B*) bearing G and C at positions 70 and 145, respectively. Panel (**B**). Heterozygous DNA (*CSN1S1B*/*CSN1S1B^bt^*) bearing two overlapping peaks at positions 70 and 145 (highlighted in yellow as S and Y, respectively), which indicate ambiguity in the attribution of G/C and C/T nucleotides, respectively.

**Table 1 foods-12-02399-t001:** Summary of the AS-PCR results for the detection of the band corresponding to the *CSN1S1B^bt^* allele in buffalo milk from single animals and bulk milk from different geographical locations. Reported are the milk samples in which the *CSN1S1B^bt^* allele was detected, with respect to the total number of samples of the same type and from the same geographical location assayed.

Milk of Single Animals from Romania	Milk of Single Animals from Bulgaria	Milk of Single Animals from the PDO Area	Bulk Milk from Romania	Bulk Milk from Bulgaria	Bulk Milk from the PDO Area
5 over 17	5 over 17	0 over 21	17 over 18	11 over 11	0 over 40

**Table 2 foods-12-02399-t002:** Comparison of methods for detecting the adulteration of Mediterranean water buffalo milk with foreign buffalo milk. Method sensitivity, analysis time, instrument cost, and analysis cost rough values are reported for dedicated methods based on a combined isoelectrofocusing-immunoblotting (IEF-IM), allele-specific polymerase chain reaction (AS-PCR) (this study), and combined isotope ratio mass spectrometry/inductively coupled plasma-mass spectrometry (IRMS/ICP-MS).

Method	Sensitivity	Analysis Time	Instrument Cost	Analysis Cost	Reference
IEF-IM	about 3% *v*/*v*	about 3 h	++	++	[5]
AS-PCR	about 0.01% *v*/*v*	about 1.5 h	+	+	this study
IRMS/ICP-MS	n.d.	about 3 h	+++++	+++	[24]

## Data Availability

The data used to support the findings of this study can be made available by the corresponding author upon request.

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
