# Peer review of "A Genotyping Method for Detecting Foreign Buffalo Material in Mozzarella di Bufala Campana Cheese Using Allele-Specific- and Single-Tube Heminested-Polymerase Chain Reaction"

_foods, 2023, doi:10.3390/foods12122399_

Round 1

Reviewer 1 Report

The work described a genotyping method to detect foreign buffalo milk in the counterpart from the PDO area. The manuscript is not well prepared, and lots of sentences are difficult to read. The method reliability should be validated with sufficient data. The advantages of cost, simplicity are not significant.

Line 18: Current methods.... should be re-written.

Line24-26: re-write this sentence.

Line 28: evaluated?

Line 29: how to achieve the simplicity, reliability, and cost?

Line 72: proven? and also the sentence should be re-written

Line 124: how many sequences were used for comparison?

Line 193: more samples are necessary for specificity test.

Line 238: where is the negative control in the agarose gel?

Line 257: it is not scientific to directly compare your LOD with others, if the pretreatment method is different.

Line 319: In addition to confirm..., should be re-written

Figures: all the agarose gels should be improved in terms of resolution.

The work described a genotyping method to detect foreign buffalo milk in the counterpart from the PDO area. The manuscript is not well prepared, and lots of sentences are difficult to read. The method reliability should be validated with sufficient data. The advantages of cost, simplicity are not significant.

Line 18: Current methods.... should be re-written.

Line24-26: re-write this sentence.

Line 28: evaluated?

Line 29: how to achieve the simplicity, reliability, and cost?

Line 72: proven? and also the sentence should be re-written

Line 124: how many sequences were used for comparison?

Line 193: more samples are necessary for specificity test.

Line 238: where is the negative control in the agarose gel?

Line 257: it is not scientific to directly compare your LOD with others, if the pretreatment method is different.

Line 319: In addition to confirm..., should be re-written

Figures: all the agarose gels should be improved in terms of resolution.

Reviewer 2 Report

The manuscript describes a protocol for evaluating if samples of buffalo mozzarella that are labeled as coming from the Campania region are actually made with buffalo milk from that region. From the data presented, it appears that the authors have largely achieved what they set out to achieve, in that they have what appears to be a relatively easy and inexpensive assay that can distinguish buffalo mozzarella that has been adulterated with buffalo milk produced in regions outside of the designated region of Campania. The assay appears to be well designed, and the paper is straight forward and well written.

One possible shortcoming of the work is that its appeal may be somewhat limited to those engaged in the production of Mozzarella di Bufala Campana (MdBC). The authors are able to take advantage of the curious fact that buffalo in the Compania region have a particular genetic allele that is different from buffalo found in other regions. I would be curious to know if the authors can think of any other examples of a PDO food that is made from animals with a particular allele that is unique to that region, i.e. if this approach would be applicable in any other cases or just this one.

I only have a few suggestions for improving the manuscript.

1. The manuscript describes two, closely-related PCR-based approaches, AS-PCR and STHN-PCR, but it was not completely clear to me from the paper which protocol the authors ultimately propose be adopted by those wishing to identify counterfeit MdBC. Is it the case that the AS-PCR protocol is preferred for assaying milk samples, but the STHN-PCR protocol is better for assaying cheese? The authors could make it more clear what the relationship is between these two protocols.

2. I think the relative low cost and ease of this approach relative to other available methods is a strong point of the paper. This point would be stronger if they put a monetary value on the different protocols shown in Table 1. What is the approximate cost to assay a sample of cheese for each of the three methods listed?

Minor point:

"required" instead of "requested" on lines 55-56.

Mostly very good. A little light editing is all that is needed. 

Round 2

Reviewer 1 Report

It seems that the authors did not modify the manuscript, according to the suggestions. For instance, the figures are still not clear. The errors with English writing must be avoided.  

It seems that the authors did not modify the manuscript, according to the suggestions. For instance, the figures are still not clear. The errors with English writing must be avoided.